# Improving Newborn Resuscitation by Making Every Birth a Learning Event

**DOI:** 10.3390/children8121194

**Published:** 2021-12-16

**Authors:** Kourtney Bettinger, Eric Mafuta, Amy Mackay, Carl Bose, Helge Myklebust, Ingunn Haug, Daniel Ishoso, Jackie Patterson

**Affiliations:** 1Department of Pediatrics, University of Kansas School of Medicine, 3901 Rainbow Blvd, MS 4004, Kansas City, KS 66103, USA; 2School of Public Health, University of Kinshasa, Kinshasa 11850, Democratic Republic of the Congo; ericmafuta2@gmail.com (E.M.); dishosok@gmail.com (D.I.); 3Department of Pediatrics, University of North Carolina at Chapel Hill, 101 Manning Drive, CB 7596, Chapel Hill, NC 27599-7596, USA; amy.mackay@unchealth.unc.edu (A.M.); carl_bose@med.unc.edu (C.B.); jackie_patterson@med.unc.edu (J.P.); 4Laerdal Medical Strategic Research Department, Tanke Svilandsgate 30, N-4002 Stavanger, Norway; Helge.Myklebust@laerdal.com (H.M.); Ingunn.Haug@laerdal.com (I.H.)

**Keywords:** resuscitation, neonatal, newborn, Helping Babies Breathe, simulation, respiratory depression, intrapartum-related mortality, debriefing

## Abstract

One third of all neonatal deaths are caused by intrapartum-related events, resulting in neonatal respiratory depression (i.e., failure to breathe at birth). Evidence-based resuscitation with stimulation, airway clearance, and positive pressure ventilation reduces mortality from respiratory depression. Improving adherence to evidence-based resuscitation is vital to preventing neonatal deaths caused by respiratory depression. Standard resuscitation training programs, combined with frequent simulation practice, have not reached their life-saving potential due to ongoing gaps in bedside performance. Complex neonatal resuscitations, such as those involving positive pressure ventilation, are relatively uncommon for any given resuscitation provider, making consistent clinical practice an unrealistic solution for improving performance. This review discusses strategies to allow every birth to act as a learning event within the context of both high- and low-resource settings. We review strategies that involve clinical-decision support during newborn resuscitation, including the visual display of a resuscitation algorithm, peer-to-peer support, expert coaching, and automated guidance. We also review strategies that involve post-event reflection after newborn resuscitation, including delivery room checklists, audits, and debriefing. Strategies that make every birth a learning event have the potential to close performance gaps in newborn resuscitation that remain after training and frequent simulation practice, and they should be prioritized for further development and evaluation.

## 1. Introduction

Each year, approximately two million pregnancies end in stillbirth and two million, and four hundred thousand newborns die within their first month after birth [1]. Over 90% of these deaths occur in low- and middle-income countries (LMICs), and most are preventable. More than one-third of neonatal deaths are attributed to events that occur during the intrapartum period [2]. A common consequence of these events is a failure of the newborn to breathe, which is referred to as respiratory depression. Under these circumstances, resuscitation to help a newborn breathe with therapies such as tactile stimulation and positive pressure ventilation (PPV) can be lifesaving. Strategies to ensure quality resuscitation are critical to reduce newborn mortality as newborn resuscitation is a key component of a bundle of interventions known to decrease intrapartum-related perinatal mortality, including increasing facility births, improving access to emergency obstetric care, and the provision of advanced neonatal care [3].

A first step in implementing quality resuscitation is to ensure that providers have appropriate knowledge and skills to help a newborn breathe. Standard newborn resuscitation training programs such as the Neonatal Resuscitation Program (NRP) [4], Neonatal Life Support (NLS) [5], and Helping Babies Breathe (HBB) [6] educate providers in resuscitation and are widely implemented on a global scale. These programs use simulation to train providers in a program-specific resuscitation algorithm informed by evidence-based practices. Standard resuscitation training increases providers’ knowledge and skills and often, but not universally, results in fewer fresh stillbirths, one-day newborn mortality, and early newborn mortality; however, there is a less consistent decrease in late neonatal mortality [7,8,9,10,11,12,13,14]. However, on-going gaps in bedside performance persist after one-time training. For example, HBB did not improve clinical practice in a rural hospital in Tanzania, and substantial performance gaps in clinical practice were noted in the Democratic Republic of the Congo (DRC) following HBB training [15,16].

A common strategy used to improve the translation of training into practice is a frequent simulation practice with a manikin [12,17]. Providers may practice complex skills such as PPV or intubation using a manikin, or they may review the steps of resuscitation with mock cases. While frequent simulation practice can mitigate declines in provider knowledge and skills and improve clinical outcomes [17,18], gaps in bedside performance may persist. For example, investigators evaluated bedside performance after HBB training with frequent simulation practice in Tanzania [19]. Among these HBB-trained providers, up to 85% of the PPV events were initiated beyond the first minute after birth, and 95% of PPV events were interrupted before one minute of continuous ventilation was administered. These gaps in quality are missed opportunities to maximize the life-saving potential of resuscitation since every 30 s delay in starting ventilation increases the risk of death or morbidity by 16% [20]. Given ongoing gaps in bedside performance despite resuscitation training with frequent simulation practice, additional interventions are required to ensure that evidence-based resuscitation reaches the bedside [21].

One of the reasons that adherence to recommended practice is so difficult is that stress may negatively affect resuscitation performance and cause the underestimation of the passage of time [22,23,24]. When interviewed, Tanzanian midwives said the stress of ventilating a non-breathing baby produced anxiety and fear, often leading to poor resuscitation performance [25]. Similarly, Norwegian midwives reported resuscitation as stressful and felt inadequate when having to bear the responsibility for both mother and newborn simultaneously [26].

One reason why newborn resuscitation is perceived as stressful is that it is a serious but relatively rare event. While population studies have shown that rates of intervention may change over time and depend on the setting, resuscitation literature consistently estimates that 5–10% of newborns receive stimulation at birth to help them breathe, while 3–6% receive PPV; fewer than 1% receive advanced resuscitation (such as medications and/or chest compressions) [27,28,29,30,31,32]. Taking into account the average number of births an individual provider attends to in one year, opportunities for clinical practice of these skills are limited. For example, by combining data from the United Nations Population Fund (UNFPA) and the World Bank, we found that there are 45 annual births per skilled birth attendant (SBA) in Tanzania, indicating these SBAs perform PPV in clinical resuscitations at an average of one to three times per year (Figure 1) [33,34]. The same sources of data suggest similar opportunities for PPV per SBA in India, whereas an SBA in the DRC performs PPV roughly 12–23 times per year. These estimates do not account for variations in delivery volume among health facilities and assumes that only one SBA participates in each resuscitation. Despite the potential flaws in these assumptions, the estimates illustrate the limited clinical opportunities to practice resuscitation, making the mastery of resuscitation skills a challenge. Although SBAs who work in large delivery units may experience a higher number of resuscitations, this does not guarantee mastery of resuscitation skills. Furthermore, the normalization of neonatal death in some facilities may reduce the stress that SBAs experience during cases that have poor outcomes.

Given the limitations of frequent simulation practice and the relatively infrequent opportunities to practice resuscitation skills during clinical care, complementary strategies that enhance performance under stress could facilitate the more effective use of lifesaving therapies at the time of birth. In this paper, we review strategies to support learning from each clinical experience that can enhance adherence to resuscitation algorithms. We broadly group these strategies into those that are used during newborn resuscitation (referred to as clinical decision support) and those that are used after newborn resuscitation (referred to as post-event reflection; Figure 2). We searched the literature for each strategy we addressed with the keywords listed in Appendix A. We also hand-combed the reference lists of relevant articles from the searches. Given the substantial burden of newborn mortality in LMICs as well as differing resuscitation environments in high- versus low-resource settings, we explored strategies to make every birth a learning event in the context of both settings.

## 2. Clinical Decision Support during Newborn Resuscitation

Clinical decision support is the provision of timely information, typically at the point of service delivery, to help providers make decisions about patient care [35]. Clinical decision support during newborn resuscitation has the potential to increase the speed of care and to mitigate the adverse effects of stress on performance. Methods for delivering clinical decision support during resuscitation must be flexible in responding to the patient’s rapidly evolving condition. They must also command the attention of the providers without distracting them from their care. In this section, we will explore four strategies to deliver clinical-decision support during newborn resuscitation: visual display of a resuscitation algorithm, peer-to-peer support, expert coaching, and automated guidance.

### 2.1. Visual Display of a Resuscitation Algorithm

Standard newborn resuscitation training programs such as NRP, NLS, and HBB include a program-specific algorithm that depicts the steps for resuscitation [36,37,38,39,40,41]. These algorithms promote memorization and a shared mental model during training [42]. After training, the algorithms are widely accessible to providers as a clinical reference in the form of printed posters, pocket guides, or increasingly digital versions (particularly in mobile health applications) [43]. In addition to serving as a resource before the point of care, these algorithms have been explored as clinical decision support tools [44].

A study of simulated resuscitations in Canada, in which participants were randomized to simulations that included an NRP algorithm poster on the wall versus simulations without the posted algorithm, demonstrated no difference in participant performance [45]. The investigators observed that the results may reflect infrequent use of the cognitive aid during the simulations, which illustrates the challenge of commanding the attention of a provider during emergency care. While HBB action plans are commonly posted in labor and delivery wards following HBB training, we are unaware of any evidence on the impact of this aid as a clinical decision support tool.

Overall, while newborn resuscitation algorithms are commonly used in training and as a clinical resource in both high- and low-resource settings, there is insufficient evidence to understand the potential utility of these algorithms as clinical decision support tools. In particular, since a visual display of the algorithm in labor and delivery rooms is frequently part of larger training interventions, the relative impact of this aid as a clinical decision support tool is difficult to assess. The limited literature on the topic suggests that the visual display of a complex algorithm may be better suited as a tool before the point of care rather than during a resuscitation.

### 2.2. Peer-to-Peer Support

Another strategy to deliver clinical-decision support during newborn resuscitation is peer-to-peer support during resuscitations attended by more than one provider. In this section, we refer to peer-to-peer support as the communication between colleagues that impacts decision-making in real-time through events such as information sharing, inquiry, assertion, shared intentions, and evaluation of plans [46]. While this may occur organically, the intentional use of this type of clinical decision support has been described in the literature as a strategy to enhance resuscitation performance.

The importance of communication on resuscitation performance is increasingly recognized but under-studied in high-resource settings. To increase the use of peer-to-peer support at the bedside, standard resuscitation training is frequently paired with team training [47]. A systematic review of team training in neonatal emergencies demonstrated that team training improves performance in simulations, and thus may improve performance at the bedside [48]. One observational analysis of team behavior during clinical newborn resuscitations in the United States illustrated the following three fundamental concepts: communication (i.e., information sharing and inquiry), management (i.e., workload and vigilance), and leadership (i.e., assertion, intentions shared, evaluation of plans) [49]. Both communication and management were variably associated with noncompliance with the NRP algorithm.

There is little evidence about peer-to-peer support during newborn resuscitations in LMICs. This may be in part because peer-to-peer support is a less accessible strategy in low-resource settings where resuscitation is commonly provided by a single birth attendant. Peer-to-peer support during resuscitation is under-studied in larger health facilities in low-resource environments. One qualitative study from Nepal indicated a willingness of peers to help (“everyone’s present and ready to help”) but also hesitation to take responsibility due to fear of individual blame (“when it comes to addressing a [resuscitation] case everybody backs out…everyone tries to escape their responsibility…they try to shove it off to one another”) [50]. These observations highlight the importance of context in the success of peer-to-peer support. A culture of improvement in a facility may be fundamental to the uptake and success of peer-to-peer support.

In summary, limited evidence suggests that peer-to-peer support may distribute workload and responsibility during a resuscitation. We hypothesize that these aspects of peer-to-peer support could reduce provider stress. While information sharing and inquiry among peers may enhance decision-making, we also hypothesize that the impact of peer-to-peer support may vary based on the resuscitation competency of those involved. In particular, qualitative literature on peer-to-peer support in low-resource contexts suggests that a facility culture that supports open communication may be critical for the uptake of this strategy. Further research is needed to understand the contextual factors that impact the effectiveness of peer-to-peer support on resuscitation performance.

### 2.3. Expert Coaching

The benefits of human support during newborn resuscitation may be enhanced if the support comes from an expert colleague rather than a peer, such as in expert coaching. Expert coaching may be delivered in person or using remote technology such as telephone or video streaming. Table 1 summarizes studies that evaluate expert coaching during neonatal resuscitations in the clinical environment.

In-person coaching by an expert has been explored as a strategy to support learners in achieving competency in newborn resuscitation in high-income countries (HICs). Investigators evaluated the impact of a resuscitation training program, including in-person coaching during neonatal resuscitation, for pediatric residents in the United States and found that this resulted in an increase in residents initiating leadership and maintaining their leadership role at low-risk deliveries [51]. Additional studies from Canada and the United States explored expert coaching as a strategy to improve PPV [52,53]. Both studies demonstrated that verbal feedback could reduce mask leak in simulation but had differing results regarding the impact of feedback on tidal volume. This preliminary work supports the need for further research on the impact of expert bedside coaching on PPV.

In-person coaching for newborn resuscitation has also been evaluated in LMICs. In a cluster-randomized trial in India, primary health centers that received training in maternal/newborn care plus in-person mentorship (including bedside coaching) had higher resuscitation knowledge than those that only received the training in maternal/newborn care [54]. In a quasi-experimental post-test with matched comparison study in primary health centers in India, birth attendants who received mentoring from nurses with a Bachelor of Science degree (including bedside coaching during deliveries) performed better on objective structured clinical examinations of newborn resuscitation than those who were not mentored [55].

To summarize, limited evidence suggests that in-person expert coaching may increase the initiation and maintenance of leadership in the delivery room, and it may also increase resuscitation knowledge and skills of providers. Since in-person expert coaching is frequently implemented as one aspect of a bundle of interventions to improve newborn resuscitation, we cannot draw conclusions about the relative contribution of this strategy to the outcomes studied. There is likely substantial variability in the way coaches provide advice at the bedside, and insufficient details are included in these studies to understand those nuances [51,52,53,54,55]. The intensity and frequency of the intervention is also unclear, so we cannot comment on how many resuscitations involved coaching or the complexity of the resuscitations for which guidance was received. The impact of expert coaching on adherence to resuscitation algorithms and neonatal outcomes has not been studied. Finally, there are constraints to the use of this strategy in LMICs where few experts in newborn resuscitation are available, and in settings where there is a low volume of births, whether in LMICs or HICs. In-person coaching by an expert external to the facility requires a substantial investment in time, particularly because the timing of complex newborn resuscitations is unpredictable. For this reason, in-person expert coaching is unlikely to provide a scalable solution in LMICs and settings where there is a low volume of births.

A potentially more scalable solution for expert coaching during newborn resuscitation is remote coaching, either by telephone or video [56,57]. Remote coaching may be used as a strategy whereby regional centers could support local providers during neonatal resuscitation. This possibility was explored in simulated neonatal resuscitations in a United States center where learners supported by a neonatologist via video consultation demonstrated decreased time to establish effective ventilation [58]. An additional study in the United States evaluated the implementation of emergency video telemedicine consultations by a referral center for high-risk newborn deliveries at lower-level hospitals [59]. The investigators found that video consultations prevented the unnecessary transfer of patients to higher levels of care and resulted in a local provider perception of improved patient safety and/or quality of care. They also noted several technical issues, reinforcing the need for a highly reliable infrastructure to provide both audio and video connections that are of sufficient quality to support this intervention. In another study, a reduction in the transfer of patients was also associated with the implementation of video telemedicine for neonatal resuscitations occurring at community hospitals in the United States [60]. A direct comparison of resuscitations supported by video telemedicine to a control group of resuscitations demonstrated higher expert rating of resuscitation quality [61]. We are unaware of any studies evaluating remote coaching during newborn resuscitation in LMICs.

Overall, limited evidence from a single center suggests that remote coaching may improve provider performance during resuscitation, but there is no evidence regarding its effect on newborn outcomes. Remote coaching may be a more scalable alternative to in-person coaching when in-facility experts are unavailable. However, the technological requirements could pose challenges for the feasibility of this strategy in low-resource settings. The effectiveness of in-person coaching versus remote coaching for newborn resuscitation has not been studied.

### 2.4. Automated Guidance

Mobile health technology can provide automated guidance at point of care as an alternative to human support. The delivery of automated guidance relies on real-time data on the newborn’s cardiorespiratory status and the ongoing care administered.

We found three studies of automated tools to support NRP providers. First, NeoCue is a proprietary software designed in the United States to aid a practitioner during newborn resuscitation with visual and auditory prompts to guide their actions, based on NRP. In a randomized study of simulated resuscitations, practitioners supported by NeoCue were more likely to perform PPV correctly (94–95% vs. 55–80%, *p* < 0.0001) and chest compressions correctly (82–93% vs. 71–81%, *p* < 0.0001), and they addressed FiO2 three times more often than the control group (*p* < 0.001) [62]. Second, a randomized simulation study in Israel evaluated an audio voice-guided application based on NRP. The investigators reported that adherence to the order sequence (100% vs. 30%), correct use of oxygen (100% vs. 25%), and performance of corrective measures for PPV (95% vs. 33%) were all better with guidance (all *p* < 0.01) [63]. Third, the NRP Prompt is a mobile health application designed to provide audiovisual prompts to providers during the resuscitation of newborns [64]. In a simulation study with residents in Canada, investigators compared performance using NRP Prompt versus visual aids only and showed no difference in performance scores or in time to PPV, intubation, or chest compressions. We are unaware of any studies evaluating these automated tools in the clinical environment.

In low-resource settings, in which few expert coaches in newborn resuscitation are available, automated guidance is a particularly attractive alternative strategy. While we found many automated tools supporting training in HBB, only two are designed to provide clinical decision support during a resuscitation [43]. NeoTap is a smartphone application that supports heart-rate assessments via auscultation by screen-tapping in order to eliminate the need for mental calculations of heart rate; it also includes an audio and visual pacer to support ventilation cadence. Investigators compared Ugandan midwives’ heart rate assessment of non-breathing newborns using NeoTap versus ECG, noting a median acquisition time of 2.7 s (IQR 1.7, 4.7) and good agreement between NeoTap and ECG for heart rate categories <60, 60–99 and ≥100 beats per minute (kappa index 0.71, 95% CI 0.63, 0.79) [65]. Liveborn is a mobile health application that is under development to provide audiovisual guidance to providers during resuscitation of newborns [66]. The application will use observer-collected data on provider actions in combination with objective newborn heart rate data from a low-cost, battery-operated heart rate meter (NeoBeat) to deliver guidance based on the HBB algorithm.

To summarize, automated guidance for clinical-decision support during newborn resuscitation is a novel possibility that has yet to be evaluated in the clinical environment. While human expert coaching can deliver more nuanced guidance than the mobile health technology currently available, artificial intelligence could enhance the effectiveness of this technology. Given the underuse of such tools in the clinical environment in both high- and low-resource settings, the ongoing exploration of automated guidance for newborn resuscitation should include research on implementation outcomes such as feasibility, acceptability, and scalability.

## 3. Post-Event Reflection after Newborn Resuscitation

Retrospectively reflecting on a challenging clinical resuscitation can be a powerful tool to improve resuscitation care. Given the relatively rare occurrence of respiratory depression for a single provider, dedicating time to learn from these events can enhance a provider’s performance for the next complex resuscitation. Furthermore, strategies that focus on learning after a bedside resuscitation can incorporate colleagues who did not actively participate in the resuscitation, thus enhancing learning for a group of providers. There are many strategies of varying complexity that can be used to support learning after a bedside resuscitation. In this section, we will review delivery room checklists, audits, case reviews, and debriefing.

### 3.1. Delivery Room Checklists

Delivery room checklists are a tool for providers to evaluate their own performance in a structured manner following a resuscitation. The checklist is a list of actions by the provider during the resuscitation filled out by those performing the resuscitation after the fact or contemporaneously by a peer functioning as a scribe. These checklists can be used to reflect on key elements of resuscitation after a clinical case. The use of a checklist can be enhanced by reviewing with a colleague such as a peer or a supervisor.

Reflection on a delivery room checklist with a peer has been evaluated as part of a larger HBB quality improvement cycle (HBB QIC) in studies in Nepal [67,68,69,70]. In HBB QIC, HBB-trained providers implemented the following quality improvement (QI) activities: self-evaluation checklists reviewed with a peer after each delivery, a daily bag-and-mask skills check, a one-day HBB refresher training at six months, and a unit progress board to monitor HBB QIC implementation (including daily statistics of number of deliveries, non-breathing infants, resuscitation cases, fresh stillbirths, neonatal deaths, and daily skills checks). In a prospective cohort study of HBB QIC, intervention resulted in a decrease in intrapartum stillbirth of 54% (aOR 0.46 [95% confidence interval (CI) 0.32, 0.66]) and a decrease in first-day neonatal mortality of 49% (aOR 0.51 [95% CI 0.31, 0.83]) [67]. In addition, a decrease in the inappropriate use of suctioning was observed (OR 0.13 [95% CI 0.09, 0.17]) and an improvement in PPV within one minute (OR 2.56 [95% CI 1.67, 3.93]). Compared to a control group, providers implementing HBB QIC retained more knowledge on a multiple-choice questionnaire six months after training (16.4 ± 1.4 vs. 12.8 ± 1.6, *p* < 0.001) [68]. A follow-up study implementing HBB QIC in a stepped-wedge cluster randomized controlled trial in Nepal found a decrease in the incidence of intrapartum-related mortality from 11.0 per 1000 births during the control period to 8.0 per 1000 births during the intervention period (aOR 0.79 [95% CI 0.69, 0.92]) [69]. Additionally, the use of PPV for babies with an Apgar score of less than seven at one minute of life increased from 3.2% to 4.0% (aOR 1.52 [95% CI 1.32, 1.77]). Providers implementing HBB QIC were also more likely to administer PPV to non-crying infants, compared to the control group (aOR 1.28 [95% CI 1.04, 1.57]) [70].

Reflection on a delivery room checklist with a supervisor has also been evaluated as part of a larger HBB QI intervention in Tanzania [71]. The HBB QI initiative began with HBB training and continued with supportive supervision visits from local expert trainers that included observations of clinical deliveries. During these visits, supervisors completed a checklist of resuscitation actions correlating to the HBB Action Plan and reviewed the checklist with the provider following the resuscitation. HBB QI increased provider knowledge on a multiple-choice questionnaire post-intervention (13.90 ± 2.02 to 15.64 ± 1.70, *p* < 0.001) and improved PPV skill performance on a manikin (6.05 ± 2.87 to 12.84 ± 1.46, *p* < 0.001). We are unaware of any studies describing delivery room checklists as an intervention in high-resource settings.

Overall, these studies demonstrate that reflection using a delivery room checklist as part of a larger QI bundle following training improves provider knowledge and skills. While there is strong evidence that QI bundles involving delivery room checklists improve newborn outcomes, the relative contribution of the checklist is likely small given the other substantial interventions in these QI bundles. Additionally, the intensity of intervention reported in these studies (i.e., how many complex resuscitations were followed by reflection with a checklist) and specific processes guiding the reflection are unclear. We hypothesize that there may be substantial variability in the quality of the reflection based on the competency of the peer or supervisor supporting the checklist review.

### 3.2. Audits

An alternative to reflecting on care with a delivery room checklist is the auditing of clinical care by an unbiased observer. In an audit, a provider who is not directly involved in the newborn’s care will review care either by direct observation or through analysis of medical records. The aim of an audit is to ensure that quality care is delivered. Newborn resuscitation audits rely heavily on direct observation since medical records provide limited detail to support a review of resuscitation care, particularly in LMICs.

Studies evaluating audits of newborn resuscitation often fall under the broader category of perinatal death audits, which involve reviewing all perinatal care provided for cases resulting in stillbirth or neonatal death [72]. Perinatal death audits have reduced perinatal mortality in LMICs including a peri-urban hospital in Uganda with a reduction of 50% [73], an urban hospital in Mozambique with a transient reduction of 50% (*p* < 0.0005) [74], and a rural health district in South Africa with a reduction of 40% (*p* = 0.002) [75]. In India and Kenya, death audits and observation of deliveries or HBB skills were included in a larger QI initiative along with frequent simulation practice of PPV, daily equipment checks, and resuscitation debriefings [76]. Overall, there was no decrease in perinatal mortality across the sites. Improvements have also been shown in HICs, including in Norway, where the countrywide perinatal mortality rate improved from 10 to 7.8 per 1000 births [77] and the Netherlands, where the nationwide term perinatal mortality rate improved from 2.3 to 2.0 per 1000 births (*p* < 0.00001) [78].

In summary, while there are multiple studies indicating that perinatal death audits reduce mortality, there is insufficient detail in these studies to understand whether improved resuscitation care is part of the causal pathway. Audits are ideal for rare events, but limited medical record documentation of delivery room care means they can be challenging to implement effectively for resuscitation care. While the use of an external party can enhance objectivity, clinical care audits also have the potential of being viewed as punitive rather than supportive of facility-based providers.

### 3.3. Debriefing

Debriefing is the practice of discussing and analyzing a clinical scenario after it occurs with the aim of improving performance in the future. The objective is to minimize the “know-do” gap in performance through dialogue between a facilitator and clinical care providers rather than lecturing or one-way feedback [79,80,81,82]. Debriefing is non-punitive, undertaken within a culture of learning, and conducted with the basic assumption that all involved are doing their best and want to improve [82]. There are several methods of debriefing, which typically share the backbone of processing emotional reactions to the event, recognizing what went well as well as what needs improvement, and identifying how to both continue the positive practices and improve other areas in future clinical care [80,81].

While debriefing is frequently integrated into simulation-based training, it can be a powerful tool to reflect on clinical care and is often underutilized [83,84,85,86]. Both NRP and HBB have recently evolved to include debriefing as an essential component of their simulation trainings [4,87,88]. HBB also encourages facilities to establish debriefing as part of their ongoing newborn resuscitation practice [88], and the NRP algorithm includes team debriefing as the final step in the resuscitation process [89]. In this section, we will focus on the literature emphasizing debriefing after bedside resuscitations. Table 2 summarizes studies that evaluate the intervention of debriefing.

In high-resource settings, oral debriefing following bedside resuscitations has been evaluated as part of a bundle of resuscitation interventions. QI intervention bundles for newborn resuscitation that include debriefing have led to improvements in both teamwork and clinical outcomes in several studies in the United States [90,91,92]. Oral debriefing following clinical resuscitations has also been evaluated as part of a bundle of interventions in three low-resource settings, in which it did not lead to overall change in perinatal deaths or fresh stillbirths [76]. Since debriefing was only evaluated within the context of a larger intervention bundle, we are unable to extrapolate what role debriefing played in these results. A scoping review on briefing and debriefing for newborn resuscitations (which included these studies [91,93]) concluded that there was insufficient new evidence to justify a systematic review or revision of resuscitation guidelines [94].

Debriefing that involves video review is a more recently developed strategy that has been touted as an improvement to more traditional oral debriefing. NRP considers a video-assisted debriefing to possess theoretical advantages over oral debriefing, although such advantages have not been proven in the setting of NRP [4]. A study conducted in Norway showed improvement in team performance and adherence to best practice guidelines for neonatal resuscitation skills after implementation of video-assisted, performance-focused debriefings following resuscitations [93]. In a single-site Australian study, investigators demonstrated that offering video recordings of newborn resuscitations to guide debriefings improved whether information was sought by the team; however, other aspects of teamwork were not improved [95]. In a study in the Netherlands and the United States, providers reported that recording and reviewing neonatal resuscitation were useful techniques, and led to learning from reviewing their own performance during resuscitation and from reviewing the performances of others [96]. Because the introduction of routine video recordings of resuscitations can be seen as overly complex, a feasibility study was conducted as part of a QI initiative in Canada. Investigators reported that video recordings of neonatal resuscitations that were used for same-day video debriefing were acceptable and implementable [97]. While we are unaware of any studies comparing video-assisted debriefings to oral debriefings after clinical resuscitations, comparative studies of the two strategies in simulation have not consistently demonstrated a significant benefit of video-assisted debriefing [98,99,100].

An exploratory study in Canada used eye-tracking glasses worn by clinicians managing the airway during neonatal resuscitations as an alternative method to guide debriefings, focused on the provider’s cognition [101]. Through ten qualitative interviews, the investigators found that the retrospective think-aloud style of debriefing prompted by the eye-tracked recordings was not only acceptable to clinicians but also a useful way to explore their cognition.

Despite the effectiveness of debriefing after newborn resuscitations, it is not routinely practiced. This may be due to several barriers to routine debriefing that vary for high- vs. low-resource settings. In HICs, perceived barriers to debriefing after neonatal resuscitations include (1) insufficient time, (2) lack of skilled facilitators, (3) lack of an appropriate setting, and (4) the threat of litigation [102]. These barriers can be addressed by (1) limiting debriefing time, using a structured approach, and postponing discussion of systems issues that require deep-dives with leadership; (2) identifying individuals with training in post-event debriefing and investing in the development of new facilitators; (3) identifying a “debriefing room” that is convenient, available at all times, and considered a safe and confidential place; (4) protecting debriefing records and discussions; and (5) developing procedures to disclose medical errors that arise during debriefings. In LMICs, barriers to debriefing often include (1) lack of prior exposure to debriefing [103,104], (2) lack of knowledge and skills to conduct debriefing [105], and (3) a culture of blame [104,106,107,108,109]. These barriers can be addressed through targeted training in debriefing, support for less-experienced facilitators, including a video review to objectively establish the sequence of events and emphasize learning points, and interventions that focus on developing a culture of improvement [104,105,106]. For example, Sim for Life Foundations is a two-day course in debriefing developed for new faculty in simulation education in Uganda [110]. A pilot study of this curriculum demonstrated that faculty who completed the training showed a significant improvement in their debriefing skills that persisted at a 12-month reassessment. A complementary strategy to debriefing training is bolstering support for less-experienced debriefing facilitators with the use of mobile health technology. The Liveborn application, described earlier in this study, also facilitates data-driven debriefing by comparing events during a resuscitation with the HBB algorithm and by supporting the provider in reflecting on their performance [66].

In both HICs and LMICs, Hofstede’s cultural analysis is commonly used to anticipate potential differences in debriefing style across different cultures, including a power distance index (PDI). A country’s PDI reflects its society’s power hierarchy between supervisors and subordinates, and countries are divided into low-PDI countries and high-PDI countries [111]. While we are unaware of studies evaluating the association between PDI and implementation of clinical debriefing, one study of PDI and simulation debriefing compared 15 high- and 11 low-PDI countries [112]. In a simulation, debriefing facilitators in high-PDI countries focused more on technical/medical skills, spoke more than the participants, asked a greater number of leading questions, and initiated more discussions. Debriefing facilitators in low-PDI countries focused more on non-technical skills, including speaking up, using closed-loop communication, system challenges, and situational awareness. These differences in the execution of debriefing in high- versus low-PDI countries could inform strategies for debriefing training. Further research is needed to understand how these differences alter the effectiveness of debriefing.

Overall, there is strong evidence from a QI study in 24 high-resource centers that debriefing, as part of a QI bundle for newborn resuscitation, improves teamwork and provider performance, especially with regard to ventilation. Debriefing following newborn resuscitation is understudied in low-resource environments, with one multi-country study suggesting no benefit in newborn outcomes. Although video-assisted debriefing is becoming more common, there is no evidence that video-assisted debriefing is more effective than oral debriefing. Barriers to routinely incorporating debriefing into clinical care exist in both HICs and LMICs, and debriefing styles differ across cultures. These differences are important to consider in developing educational programming using debriefing.

## 4. Conclusions

Improving adherence to resuscitation algorithms is vital to preventing neonatal deaths as a result of respiratory depression. Standard resuscitation training programs combined with frequent simulation practice have not reached their life-saving potential due to ongoing gaps in bedside performance. Complex neonatal resuscitations, such as those involving PPV, are relatively uncommon for any given resuscitation provider, making consistent clinical practice an unrealistic solution for improving performance. Bedside learning strategies such as clinical-decision support and post-event reflection can maximize this limited potential for clinical practice. Expert coaching is a promising strategy for clinical-decision support in high-resource settings with high births and requires further study to understand its effect on clinical outcomes. Novel methods to deliver expert guidance such as remote coaching or automated guidance warrant further investigation, particularly for settings where in-person expert coaching is not scalable. Debriefing is an effective strategy for post-event reflection that improves provider performance. Implementation strategies that address barriers to debriefing may be critical for the increased uptake of this strategy in both high- and low-resource environments. Strategies that make every birth a learning event have the potential to close the performance gaps in newborn resuscitation that remain after training and frequent simulation practice, and they should be prioritized for further development and evaluation.

## Figures and Tables

**Figure 1 children-08-01194-f001:**
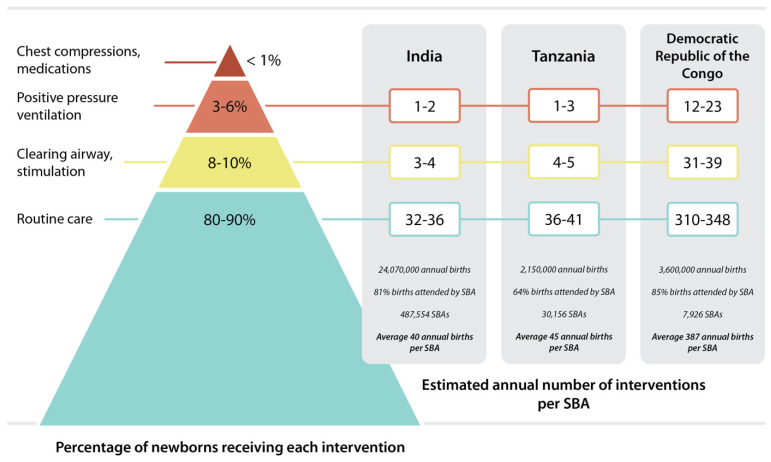
Estimated annual resuscitation interventions per skilled birth attendant (SBA). Percentage of newborns receiving each intervention derived from Lee et al. BMC Public Health 2011 [27]. Annual births reflect 2020 data derived from Our World in Data (https://ourworldindata.org/grapher/annual-number-of-births-by-world-region?tab=table&time=earliest, accessed on 23 September 2021). Percent births attended by a SBA derived from the most current UNICEF/WHO data per country (2016 for India and DRC; 2018 for Tanzania; https://data.unicef.org/topic/maternal-health/delivery-care/, accessed on 23 September 2021). Number of SBAs per country derived from current trajectory for 2020 from the UNFPA midwifery dashboard (https://www.unfpa.org/data/sowmy/TZ, accessed on 23 September 2021). Estimated annual number of interventions per SBA reflects the assumption that each birth is attended by only one SBA. The average number of interventions per SBA will vary based on the delivery census at their facility.

**Figure 2 children-08-01194-f002:**
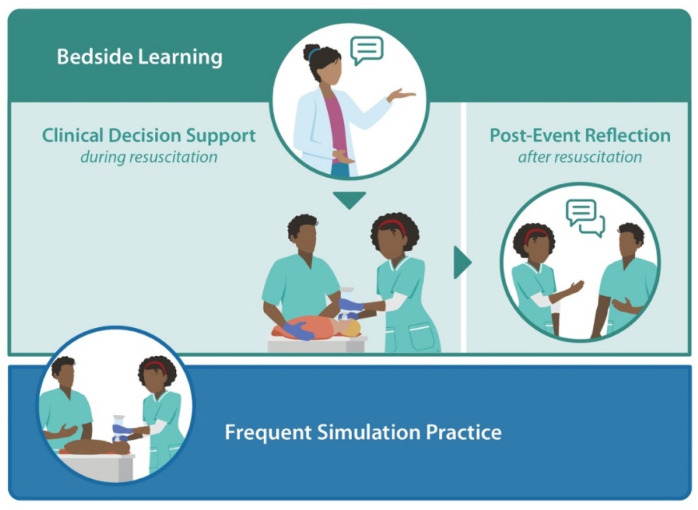
Strategies for improving the translation of learning into practice: frequent simulation practice, real-time guidance, and debriefing.

**Table 1 children-08-01194-t001:** Studies evaluating expert coaching during clinical neonatal resuscitations.

Reference	Type of Study	Location	Objective	Intervention (Control)	Outcome(s)
**In-Person Coaching**					
[51]	Observational study	USA	To evaluate the impact of a resuscitation training program for pediatric residents on teamwork, communication, and resident leadership in the delivery room	In-person coaching during neonatal resuscitation by advanced providers (neonatologists, neonatal fellows, or nurse practitioners) as well as didactic teaching, simulation training, and review of video recordings of clinical resuscitations	Increase in resident initiating leadership at low-risk deliveries (31% vs. 93%, *p* < 0.001) Increase in maintaining leadership role throughout resuscitation (19% vs. 79%, *p* < 0.001)
[54]	Cluster randomized trial	India	To evaluate the effectiveness of mentorship in improving quality of care of births in primary health centers	Maternal/newborn care training plus in-person mentorship from nurse midwives including bedside coaching, case demonstrations and job-aids (vs. maternal/newborn care training only)	Increase in resuscitation knowledge (aOR 10.7 [95% CI 4.6, 25.0])
[55]	Quasi-experimental post-test with matched comparison study	India	To assess whether mentorship improved quality of care provided by birth attendants during childbirth	Mentoring from nurses with a Bachelor of Science degree with bedside coaching during normal and complicated deliveries, as well as didactic instruction on maternal/newborn care, including respiratory depression (vs. no mentoring)	Improved performance on objective structured clinical examinations of newborn resuscitation (28.4% increase of 95% CI 23.2, 33.7)
**Remote Coaching**					
[59]	Observational study	USA	To evaluate the implementation of video telemedicine consultations by a referral center for high-risk newborn deliveries at lower-level hospitals	Telemedicine consultations by neonatologists for prematurity, respiratory distress, and need for advanced resuscitations	Prevented unnecessary transfer of patients to higher levels of care Improved patient safety and/or quality of care per local provider perception
[60]	Multiple-baseline study	USA	To evaluate the effect of video telemedicine for neonatal resuscitations on the transfer of newborns from community hospitals to facilities with advanced newborn intensive care units	Video telemedicine for neonatal resuscitations occurring at community hospitals	Decrease in transfers (by 0.70 transfers per facility-month, 95% CI −1.236, −0.157)
[61]	Retrospective cohort study	USA	To compare newborns who experienced resuscitations with video telemedicine to those who experienced resuscitations without video telemedicine	Video telemedicine for neonatal resuscitations occurring at community hospitals	Higher expert rating of resuscitation quality (expert rating score, range 1–10, 10 = no room for improvement; intervention group median of 7 [IQR 3, 8] versus control of 4 [IQR 3, 5], *p* = 0.002) No difference in percent of newborns with a heart rate > 100 at five or 10 min, nor successful intubations

**Table 2 children-08-01194-t002:** Studies evaluating debriefing after clinical neonatal resuscitation.

Reference	Type of Study	Location	Objective	Intervention (Control)	Outcome(s)
[90]	Quality improvement	USA	To improve teamwork and quality of care during neonatal resuscitation	Readiness Bundle (including pre-briefing, an equipment preparation checklist, and debriefing) implemented as part of a delivery room QI collaborative	31% of NICUs identified debriefing as the most effective component of the Readiness Bundle 100% of NICUs would recommend the bundle to other NICUs
[91]	Quality improvement	USA	To improve teamwork and quality of care during resuscitations of potentially high-risk infants	High-risk delivery checklist including equipment preparation, pre-briefing, and debriefing	Decrease in percentage of resuscitations with communication problems (23% vs. 4%, *p* < 0.01) No change in equipment preparation/use (21% vs. 23%), inappropriate decisions (33% vs. 27%), leadership (21% vs. 18%), procedure sequence/timing/technique (10% vs. 6%)
[92]	Quality improvement comparison study	USA	To evaluate the effectiveness of a perinatal collaborative quality improvement initiative compared to independent local initiatives	Readiness Bundle (including pre-briefing, an equipment preparation checklist and debriefing) implemented as part of a delivery room QI collaborative (vs standard care outside the QI collaborative)	Decrease in hypothermia (39% to 21%, *p* < 0.0001), intubations (53% to 40%, *p* < 0.0001), surfactant use in the delivery room (37% to 20%, *p* < 0.0001) Increase in hyperthermia (6% to 9%, *p* = 0.0015), CPAP use (aOR 1.65 [95% CI 1.42, 1.93]), CPAP use without intubation [aOR 2.40 [95% CI 2.05, 2.85])
[76]	Pre-post study	India and Kenya	To evaluate implementation of an HBB intervention bundle at three sites in LMIC	Intervention bundle of debriefing, death audits, observation of deliveries or HBB skills, frequent simulation practice of PPV, and daily equipment checks	No change in perinatal deaths (estimate of pre-post differences in mortality rates 2.34 (95% CI −3.11, 7.80)), fresh stillbirths (estimate of pre-post differences in mortality rates 3.75 (95% CI −0.21, 7.70)) In newborns >1500 gm in Kenya: decrease in perinatal death (38.5 to 28.2 per 1000 births, *p* = 0.03), fresh stillbirth (25.7 to 16.4 per 1000 births, *p* = 0.03)
[93]	Pre-post study	Norway	To assess teamwork and quality of care during neonatal resuscitations before and after implementation of debriefings with video recordings	Debriefings supported by video-recorded resuscitations and led by two experienced facilitators, focusing on guideline adherence and non-technical skills	Increase in team performance (88% to 100%, *p* < 0.001), delivery of adequate PPV (70% to 100%, *p* < 0.001) Decrease in pauses during initial ventilation (20% to 0%, *p* = 0.02) No change in proportion of infants with heart rate > 100 bpm at 2 min (71% to 82%, *p* = 0.22)
[95]	Pre-post study	Australia	To evaluate teamwork and quality of care during neonatal resuscitations before and after implementation of debriefings with video recordings	Debriefings (including the video recordings of resuscitations) at set periods of time to clinicians who chose voluntarily to attend	Increase in information seeking No change in activity coordination, information sharing, problem identification, team member support, effective use of human resources, planning, discussion, anticipation of next steps
[96]	Qualitative study	The Netherlands and USA	To examine providers’ perception of video recording and reviewing neonatal resuscitations	Recording and reviewing of neonatal resuscitations	Intervention was useful, improved time perception, reflection on guideline compliance, and acted less invasively during resuscitations
[97]	Prospective cohort quality improvement study	Canada	To evaluate the feasibility of video recording during neonatal resuscitations	QI package including pre-resuscitation team huddle and same-day debriefing supported by video recording of resuscitation	Intervention was acceptable and implementable

Abbreviations: QI, quality improvement; NICU, neonatal intensive care unit; HBB, Helping Babies Breathe; LMIC, low- and middle-income countries; CI, confidence interval

## Data Availability

Data sharing not applicable. No new data were created or analyzed in this study as it is a review.

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
