# Peer review of "Improving Newborn Resuscitation by Making Every Birth a Learning Event"

_children, 2021, doi:10.3390/children8121194_

Round 1

Reviewer 1 Report

Well written manuscript. It would be great if a table could be made for  better representation of data available from previous research. 

additionally, the keyword used to search the literature would increase the validity and reliability of the search.

Moreover, one of the free-of-charge smartphone application, NeoTap, which records heart rate with a screen-tapping method, avoiding mental arithmetic calculations, could have been mentioned. The prospective observational study reference is as below:

Myrnerts Höök S, Pejovic NJ, Cavallin F, et al. Smartphone app for neonatal
heart rate assessment: an observational study. BMJ Paediatrics Open
2020;4:e000688. doi:10.1136/bmjpo-2020-000688

Being a free app, low and middle income countries could use it.

Author Response

Point 1: It would be great if a table could be made for better representation of data available from previous research. 

Response 1: We have added Tables 1 and 2 to the manuscript to summarize the data available for expert coaching and debriefing. We selected these two strategies for the creation of tables given the larger amount of studies reviewed in these sections compared to the other strategies covered in this manuscript.

Point 2: additionally, the keyword used to search the literature would increase the validity and reliability of the search.

Response 2: Each author conducted an independent search for the section that they wrote. We have compiled these search strategies into Supplemental Table 1 and included a couple of sentences referring to this table on lines 119-121. This new addition reads as follows: “We searched the literature for each strategy we addressed with the keywords listed in Supplemental Table 1. We also hand-combed reference lists of relevant articles from the searches.”

Point 3: Moreover, one of the free-of-charge smartphone application, NeoTap, which records heart rate with a screen-tapping method, avoiding mental arithmetic calculations, could have been mentioned. The prospective observational study reference is as below:

Myrnerts Höök S, Pejovic NJ, Cavallin F, et al. Smartphone app for neonatal
heart rate assessment: an observational study. BMJ Paediatrics Open
2020;4:e000688. doi:10.1136/bmjpo-2020-000688

Being a free app, low and middle income countries could use it.

Response 3: We have added a description of NeoTap to the automated guidance section on lines 451-458. This new addition reads as follow: “NeoTap is a smartphone application which supports heart rate assessment via auscultation by screen-tapping in order to eliminate the need for mental calculations of heart rate; it also includes an audio and visual pacer to support ventilation cadence. Investigators compared Ugandan midwives’ heart rate assessment of non-breathing newborns using NeoTap versus ECG, noting a median acquisition time of 2.7 seconds (IQR 1.7, 4.7) and good agreement between NeoTap and ECG for heart rate categories <60, 60-99 and ≥100 beats per minute (kappa index 0.71, 95% CI 0.63, 0.79).”

Reviewer 2 Report

This paper is a narrative review aimed at identifying strategies to support learning by care providers in the context of newborn resuscitation. The areas of focus are the use of clinical decision supports that can be used during resuscitation, such as flow diagrams and other types of cognitive aids, peer communication, or trained/skilled observers who provide prompts, and methods used after the resuscitation including the use of debriefing or audit using checklists, provider recall, documentation or video recording. Considerations for both high and low resource settings are discussed. 

The paper covers these subjects comprehensively and clearly, and I have not identified any major omissions or inappropriate self-citation.  The review is topical and the paper summarises both available evidence and gaps in knowledge. I am not aware of other review articles covering the same scope of subject matter (although component topics have been addressed elsewhere). 

Specific Comments:

  1. In the first paragraph - general comments are made about worldwide stillbirths and neonatal deaths, and the proportion of these deaths that may be related to intrapartum events. The numerators, denominators and sources of data for the various estimates are different and the reader is left without any accurate sense of the number of lives that could be saved by skilled resuscitation. Can such an estimate be more accurately derived, or would it be better to emphasise that neonatal resuscitation is a key component of a bundle of interventions that include better access to and quality of antenatal, intrapartum and newborn care?
  2. Page 2 paragraph beginning line 79. There is an assumption that because babies receive various resuscitation interventions, they need them. Reference 24 does not provide original data for either the number receiving or requiring resuscitation, but cites estimates from other review articles and textbooks. Some population-based data are available on the number of infants who receive various resuscitation interventions. For example, the Australian Institute of Health and Welfare publish such data, and a recent paper (Kapadia et al. Trends in neonatal resuscitation patterns in Queensland, Australia - A 10-year retrospective cohort study. Resuscitation. 2020 Dec;157:126-132 provides data for one Australian state. There are also some reports from very large, single hospital databases. It is apparent from these sources that the population rates of interventions can change over time and by location, and may or may not reflect "requirement". The need may be higher or lower than the population rates depending on location. There is also a general assumption in newborn resuscitation that the better the earlier steps are performed, the fewer advanced measures are needed. Overall, I realise that the point of the paragraph is to illustrate that wide variation in case load and case mix for providers can mean that some get a lot more clinical experience than others. However, I suggest rewriting the paragraph to use actual data, where available. There are a few other places in the manuscript where it would be helpful to change from "requiring" resuscitation to "receiving" resuscitation. 
  3. Page 3 - Section on visual display of a resuscitation algorithm. I think many providers would suggest that flow diagrams and visual graphics may be used more before the point of care, (such as in training and post-training revision) to illustrate and help with memorisation of a shared mental model for the resuscitation. Using them during the resuscitation, when providers need to give visual and cognitive attention to the baby, to monitors if available, and to performance of what can be quite challenging interventions may not be their best and highest use. Is it reasonable to discuss that their potential may lie elsewhere? 
  4. Page 3 line 44. It would help to have definitions and perhaps tangible examples of "peer to peer support" and "team-based resuscitation" as these terms may be open to a variety of interpretations. 
  5. Page 7 - It would be useful to have a definition or example of what is meant by a "delivery room checklist". Many readers may think of this as a checklist for equipment set up, team briefing etc. Do the authors mean it as a "resuscitation record sheet", filled out by those performing the resuscitation (after the fact) or contemporaneously by a person assigned as scribe? 
  6. It would be helpful to have some indication from the authors as to their appraisal of the certainty of evidence they cite. While a full risk of bias and certainty of evidence assessment would require extensive work and be beyond the scope of the paper, some comments about adequacy of the evidence cited would markedly improve the paper, and help readers.
  7. Quite a bit of the evidence in various sections would be better summarised in tables, with a shorter narrative. While the paper is clearly written, the text is lengthy and dense. The tables could also include sample sizes (some are very small) and other strengths and drawbacks of the studies that are discussed. 

Author Response

Improving Newborn Resuscitation by Making Every Birth a Learning Event

Kourtney Bettinger, Eric Mufata, Amy Mackay, Carl Bose, Helge Myklebust, Ingunn Haug, Daniel Ishoso, Jackie Patterson

Point 1: In the first paragraph - general comments are made about worldwide stillbirths and neonatal deaths, and the proportion of these deaths that may be related to intrapartum events. The numerators, denominators and sources of data for the various estimates are different and the reader is left without any accurate sense of the number of lives that could be saved by skilled resuscitation. Can such an estimate be more accurately derived, or would it be better to emphasise that neonatal resuscitation is a key component of a bundle of interventions that include better access to and quality of antenatal, intrapartum and newborn care?

Response 1: We have added a phrase contextualizing newborn resuscitation as a key component of a bundle of interventions on lines 45-49. This new addition reads as follows: “Strategies to ensure quality resuscitation are critical to reduce newborn mortality as newborn resuscitation is a key component of a bundle of interventions known to decrease intrapartum-related neonatal mortality including increasing facility births, improving access to emergency obstetric care, and provision of advanced neonatal care.”

Point 2: Page 2 paragraph beginning line 79. There is an assumption that because babies receive various resuscitation interventions, they need them. Reference 24 does not provide original data for either the number receiving or requiring resuscitation, but cites estimates from other review articles and textbooks. Some population-based data are available on the number of infants who receive various resuscitation interventions. For example, the Australian Institute of Health and Welfare publish such data, and a recent paper (Kapadia et al. Trends in neonatal resuscitation patterns in Queensland, Australia - A 10-year retrospective cohort study. Resuscitation. 2020 Dec;157:126-132 provides data for one Australian state. There are also some reports from very large, single hospital databases. It is apparent from these sources that the population rates of interventions can change over time and by location, and may or may not reflect "requirement". The need may be higher or lower than the population rates depending on location. There is also a general assumption in newborn resuscitation that the better the earlier steps are performed, the fewer advanced measures are needed. Overall, I realise that the point of the paragraph is to illustrate that wide variation in case load and case mix for providers can mean that some get a lot more clinical experience than others. However, I suggest rewriting the paragraph to use actual data, where available. There are a few other places in the manuscript where it would be helpful to change from "requiring" resuscitation to "receiving" resuscitation. 

Response 2: We have added several references of population-based rates or large, single hospital databases to support the estimates of newborns receiving various interventions. As these studies noted estimates that largely fell within the range from reference 24, we have elected to maintain these ranges as a global estimate for the purposes of Figure 1. We also added the caveat that rates of intervention may change over time and depend on the setting. We changed the word “requiring” to “receiving” or “involving” throughout the manuscript.

Point 3: Page 3 - Section on visual display of a resuscitation algorithm. I think many providers would suggest that flow diagrams and visual graphics may be used more before the point of care, (such as in training and post-training revision) to illustrate and help with memorisation of a shared mental model for the resuscitation. Using them during the resuscitation, when providers need to give visual and cognitive attention to the baby, to monitors if available, and to performance of what can be quite challenging interventions may not be their best and highest use. Is it reasonable to discuss that their potential may lie elsewhere? 

Response 3: We revised this section of the manuscript to highlight the typical use of these algorithms (for memorization and shared mental models during training). We also emphasized their utility in training and as a tool before the point of care rather than as a clinical decision support tool.

Point 4: Page 3 line 44. It would help to have definitions and perhaps tangible examples of "peer to peer support" and "team-based resuscitation" as these terms may be open to a variety of interpretations. 

Response 4: To minimize terminology and confusion, we deleted references to team-based resuscitation and provided a definition of peer-to-peer support on lines 220-223. This new addition reads as follows: “In this section, we refer to peer-to-peer support as communication between colleagues that impacts decision-making in real-time through events such as information sharing, inquiry, assertion, shared intentions, and evaluation of plans.”

Point 5: Page 7 - It would be useful to have a definition or example of what is meant by a "delivery room checklist". Many readers may think of this as a checklist for equipment set up, team briefing etc. Do the authors mean it as a "resuscitation record sheet", filled out by those performing the resuscitation (after the fact) or contemporaneously by a person assigned as scribe? 

Response 5: We have added a definition of a delivery room checklist on lines 486-488. This new addition reads as follows: “The checklist is a list of actions by the provider during the resuscitation filled out by those performing the resuscitation after the fact or contemporaneously by a peer functioning as a scribe.”

Point 6: It would be helpful to have some indication from the authors as to their appraisal of the certainty of evidence they cite. While a full risk of bias and certainty of evidence assessment would require extensive work and be beyond the scope of the paper, some comments about adequacy of the evidence cited would markedly improve the paper, and help readers.

Response 6: In the summary paragraphs at the end of each section, we edited our language to better emphasize our appraisal of the certainty of the evidence cited. These summary paragraphs start on the following lines: 153, 248, 305, 340, 463, 526, 568, and 800.

Point 7: Quite a bit of the evidence in various sections would be better summarised in tables, with a shorter narrative. While the paper is clearly written, the text is lengthy and dense. The tables could also include sample sizes (some are very small) and other strengths and drawbacks of the studies that are discussed. 

Response 7: We have added Tables 1 and 2 to the manuscript to summarize the data available for expert coaching and debriefing. We selected these two strategies for the creation of tables given the larger amount of studies reviewed in these sections compared to the other strategies covered in this manuscript. Given these tables, we have decreased the amount of text in these sections accordingly.

Reviewer 3 Report

Attached comments

Author Response

Improving Newborn Resuscitation by Making Every Birth a Learning Event

Kourtney Bettinger, Eric Mufata, Amy Mackay, Carl Bose, Helge Myklebust, Ingunn Haug, Daniel Ishoso, Jackie Patterson

Point 1: Line 52-53  This statement needs to be nuanced since interventions have provided mixed results (ref 67 in present manuscript). Most interventions have had god results on early neonatal mortality rate (0-7 days), but not on overall neonatal mortality (0-28 days). This problem was also highlighted in the study:  Improved postnatal care is needed to maintain gains in neonatal survival after the implementation of the Helping Babies Breathe initiative Acta Paediatr. J Wrammert, A KC, U Ewald, M Målqvist

Response 1: We added more language to reflect the nuances of outcomes following resuscitation training on lines 57-59. The text with this new addition reads as follows: “Standard resuscitation training increases providers’ knowledge and skills and often, but not universally, results in decreased fresh stillbirths, one-day newborn mortality, and early newborn mortality. There is less consistently a decrease in late neonatal mortality.”

Point 2: I find the assumptions in figure 1 (line 87) somewhat problematic (as you have also mentioned). The variation within a country may be extremely high as you have suggested. SBA in LMIC that work in large delivery units may be exposed to a very high number of resuscitations without achieving any form of mastery. The level of stress may be very low because bad outcome is normalized. Members of our research group are soon to publish qualitative data on this. Can this problem be highlighted in any way?

Response 2: We added two sentences to highlight that more resuscitations does not guarantee mastery of skills and that normalization of death may reduce stress experienced with poor outcomes on lines 99-111. This new addition reads as follows: “Although SBAs working in large delivery units may experience a higher number of resuscitations, this does not guarantee mastery of resuscitation skills. Furthermore, normalization of neonatal death in some facilities may reduce the stress that SBAs experience during cases with poor outcomes.”

Point 3: Lines 269-278 No references are given for the two cited studies

Response 3: We inserted the references for the two cited studies in lines438 and 442.

Point 4: Automated guidance (line 264)  Several studies have been published on the NeoTap Mhealth resuscitation support app. This system has been developed by our own study group, so I am well aware that this comment is biased. However, it is the only system tailored for low resource settings that is entirely free of charge and readily available for all users worldwide in both English and French for both IOS and Android. It provides ventilation support with an audio and visual pacer, heart rate monitoring and audio guidance. 5 papers have evaluated various versions of the application. The latest published paper is: Smartphone app for neonatal heart rate assessment: an observational study, BMJ Pediatrics 2020. Since this comment reflects our own work, feel free to dismiss it.

Response 4: We have added a description of NeoTap to the automated guidance section on lines 451-458. This new addition reads as follows: “NeoTap is a smartphone application which supports heart rate assessment via auscultation by screen-tapping in order to eliminate the need for mental calculations of heart rate; it also includes an audio and visual pacer to support ventilation cadence. Investigators compared Ugandan midwives’ heart rate assessment of non-breathing newborns using NeoTap versus ECG, noting a median acquisition time of 2.7 seconds (IQR 1.7, 4.7) and good agreement between NeoTap and ECG for heart rate categories <60, 60-99 and ≥100 beats per minute (kappa index 0.71, 95% CI 0.63, 0.79).”

Point 5: Debriefing (lines 463-464)  In fact, this study in the context of HBB training, looked at the effects of video-debriefing in a low-resource setting Adding video-debriefing to Helping-Babies-Breathe training enhanced retention of neonatal resuscitation knowledge and skills among health workers in Uganda: a cluster randomized trial, Global Health Actions, 2020

Response 5: We adjusted the wording of the sentence on lines 713-716 to include this study in Uganda and added it to the references cited at the end of the sentence. This adjustment reads as follows: “While we are unaware of any studies comparing video-assisted debriefing to oral debriefing after clinical resuscitations, comparative studies of the two strategies in simulation have not consistently shown significant benefit of video-assisted debriefing.”

Round 2

Reviewer 2 Report

Thanks for comprehensively considering the points raised in review. Each point has been addressed clearly. I have one additional question about one of the new amendments:

p 3 Line 102; The new sentence; "Furthermore, normalization of neonatal death in some facilities may reduce the stress that SBAs experience during cases with poor outcomes" seems ambiguous. Is the "with" meant to indicate "thus resulting in" (causation) or "that have", without any suggestion of causation? In other words, is the intended meaning that in facilities that consistently produce poor outcomes, deaths may not induce the same stress in birth attendants as in locations where death is uncommon (which might seem a good thing, in view of the points raised in the previous paragraph), or is the point more that a certain level of stress is needed to encourage improved performance, and this might be lacking where practitioners get used to death? This would seem to conflict with the preceding paragraph, where the potential for stress to impair performance is raised, although it might still be true. There could be a "Goldilocks" phenomenon with stress, where some is needed to motivate training and performance, but too much could reduce motor performance, inhibit decision-making or induce reluctance to participate in a team. For example, there are mixed effects of beta adrenergic blockade on surgical performance.  Perhaps rewording the sentence or splitting it might help elaborate the authors' main points. 

Author Response

Point 1: 

Thanks for comprehensively considering the points raised in review. Each point has been addressed clearly. I have one additional question about one of the new amendments:

p 3 Line 102; The new sentence; "Furthermore, normalization of neonatal death in some facilities may reduce the stress that SBAs experience during cases with poor outcomes" seems ambiguous. Is the "with" meant to indicate "thus resulting in" (causation) or "that have", without any suggestion of causation? In other words, is the intended meaning that in facilities that consistently produce poor outcomes, deaths may not induce the same stress in birth attendants as in locations where death is uncommon (which might seem a good thing, in view of the points raised in the previous paragraph), or is the point more that a certain level of stress is needed to encourage improved performance, and this might be lacking where practitioners get used to death? This would seem to conflict with the preceding paragraph, where the potential for stress to impair performance is raised, although it might still be true. There could be a "Goldilocks" phenomenon with stress, where some is needed to motivate training and performance, but too much could reduce motor performance, inhibit decision-making or induce reluctance to participate in a team. For example, there are mixed effects of beta adrenergic blockade on surgical performance.  Perhaps rewording the sentence or splitting it might help elaborate the authors' main points. 

Response 1: Thank you for this thoughtful feedback. We have updated the sentence as you recommend in lines 109-111. This sentence now reads as follows: "Furthermore, normalization of neonatal death in some facilities may reduce the stress that SBAs experience during cases that have poor outcomes."